# Nicotinamide riboside for peripheral artery disease: the NICE randomized clinical trial

Mary M. McDermott [1,2] ✉, Christopher R. Martens [3], Kathryn J. Domanchuk[1], Dongxue Zhang[1], Clara B. Peek [1,4], Michael H. Criqui[5], Luigi Ferrucci [6], Philip Greenland[1,2], Jack M. Guralnik[7], Karen J. Ho[8], Melina R. Kibbe[9], Kate Kosmac[10], Donald Lloyd-Jones[1,2], Charlotte A. Peterson [11], Robert Sufit[12], Lu Tian[13], Stephanie Wohlgemuth[14], Lihui Zhao[2], Pei Zhu[1,4] & Christiaan Leeuwenburgh[14]

People with lower extremity peripheral artery disease (PAD) have increased oxidative stress, impaired mitochondrial activity, and poor walking performance. NAD+ reduces oxidative stress and is an essential cofactor for mitochondrial respiration. Oral nicotinamide riboside (NR) increases bioavailability of NAD+ in humans. Among 90 people with PAD, this randomized double-blind clinical trial assessed whether 6-months of NR, with and without resveratrol, improves 6-min walk distance, compared to placebo, at 6-month follow-up. At 6-month follow-up, compared to placebo, NR significantly improved 6-min walk (+7.0 vs. −10.6 meters, between group difference: +17.6 (90% CI: + 1.8,+∞). Among participants who took at least 75% of study pills, compared to placebo, NR improved 6-min walk by 31.0 meters and NR + resveratrol improved 6-min walk by 26.9 meters. In this work, NR meaningfully improved 6-min walk, and resveratrol did not add benefit to NR alone in PAD. A larger clinical trial to confirm these findings is needed. Clinical Trials.gov registration: NCT03743636.

People with lower extremity peripheral artery disease (PAD) have severe walking disability, but few effective treatments exist[1]. In PAD, lower extremity ischemia causes insufficient oxygen and nutrient delivery to lower extremity skeletal muscle, which increases oxidative stress, damages skeletal muscle fibers, and impairs mitochondria function.[2–7]

NAD+ is an essential coenzyme for mitochondrial respiration and a co-substrate for enzymes involved in metabolic regulation, cellular stress resistance, and DNA damage repair, such as poly (ADP-ribose) polymerase (PARP)[8,9]. In preclinical study, NAD+ activated endothelial nitric oxide synthase (eNOS) and reduced oxidative stress to increase nitric oxide abundance[10]. Greater NAD+ abundance increased sirtuin1 (SIRT1) expression and improved skeletal muscle health, mitochondrial activity, and nitric-oxide-mediated endothelial function[11–13]. Nicotinamide riboside (NR) is a precursor to NAD+ and is available as an over-the counter supplement. In 2022, sales of NR in the U.S.

[1]Northwestern University Feinberg School of Medicine, Department of Medicine, Chicago, IL, USA. [2]Northwestern University Feinberg School of Medicine, Department of Preventive Medicine, Chicago, IL, USA. [3]University of Delaware, Department of Kinesiology & Applied Physiology, Newark, DE, USA. [4]Northwestern University Feinberg School of Medicine, Department of Biochemistry and Molecular Genetics, Chicago, IL, USA. [5]University of California at San Diego, Division of Preventive Medicine, San Diego, CA, USA. [6]National Institute on Aging, Division of Intramural Research, Baltimore, MD, USA. [7]University of Maryland School of Medicine, Department of Epidemiology and Public Health, Baltimore, MD, USA. [8]Northwestern University Feinberg School of Medicine, Department of Surgery, Chicago, IL, USA. [9]University of Virginia, Department of Surgery, Charlottesville, VA, USA. [10]Augusta University, Department of Physical Therapy, Augusta, GA, USA. [11]University of Kentucky, Center for Muscle Biology, Lexington, KY, USA. [12]Northwestern University Feinberg School of Medicine, Department of Neurology, Chicago, IL, USA. [13]Stanford University, Department of Health Research and Policy, Palo Alto, CA, USA. [14]University of Florida, Department of Physiology and Aging, Gainesville, FL, USA. ✉e-mail: mdm608@northwestern.edu

exceeded 60 million dollars. In mice, oral NR, a precursor to NAD+, increased skeletal muscle SIRT-1 and mitochondrial activity and improved limb strength and running endurance[13,14]. In humans, oral NR increased NAD+ abundance[15,16].

Resveratrol, a naturally occurring polyphenol, may increase SIRT1 affinity for NAD+[17]. Therefore, the **NIC**otinamid**E** riboside with and without resveratrol to improve functioning in PAD (**NICE**) clinical trial tested the hypotheses that NR alone improves 6-min walk distance, compared to placebo, and that NR + resveratrol improves 6-min walk distance, compared to placebo, in people with PAD.

Here we show that NR meaningfully improves 6-min walk, and resveratrol does not add benefit to NR alone in PAD.

## Results

Ninety participants were randomized and 89 (98.9%) completed 6-month follow-up (Fig. 1). Of 104 people who began run-in, eight (7.7%) did not meet criteria for passing and were excluded (Supplementary Table 1). Of 90 randomized, mean age was 71.2 years (standard deviation (SD): 9.1), 47.8% were Black, and 46.7% were female (Table 1). Proportions of participants with at least 75% adherence to pills were 75% for NR, 52% for NR + resveratrol, and 76% for placebo.

### Primary outcomes

Compared to placebo, NR improved 6-min walk by 17.6 meters (90% CI: +1.77, +∞, $P = 0.08$) at 6-month follow-up, meeting the pre-specified criterion for statistical significance (Table 2 and Fig. 2A). Compared to placebo, NR+ resveratrol did not significantly improve 6-min walk at 6-month follow-up (+3.65 meters (90% CI:−11.2, +∞, $P = 0.38$)) (Table 2 and Fig. 2A).

### Secondary outcomes

At 3-month follow-up, compared to placebo, NR alone (+22.4 meters, (90% CI: +7.3,+∞), $P = 0.029$) and NR + resveratrol (+20.6 meters (90% CI: +6.3,+∞), $P = 0.034$) each significantly improved 6-min walk (Table 2, Fig. 2B). At 6-month follow-up, compared to placebo, NR significantly improved peak treadmill walking time (+2.1 min (90% CI: +0.24, +∞, $P = 0.08$)), while NR + resveratrol did not significantly improve treadmill walking time (+1.7 min (90% CI:−0.21, +∞, $P = 0.12$)) (Table 3). At 6-month follow-up, compared to placebo, NR alone did not significantly improve the WIQ distance score (−7.1 (90% CI:−15.2, +∞, $P = 0.87$)), physical activity total counts/day (+10,995 (90% CI:−2,078, +∞, $P = 0.14$)), or physical activity counts/min (+8.24 (90% CI:−23.85, +∞, $P = 0.37$)) (Table 3). At 6-month follow-up, compared to placebo, NR + resveratrol did not significantly improve the WIQ distance score (−5.1 (90% CI:−12.6, +∞, $P = 0.80$)), physical activity total counts/day (−842 (90% CI:−13,125, +∞, $P = 0.54$)), or physical activity counts/min (+7.47 (90% CI:−22.97, +∞, $P = 0.38$)) (Table 3).

Compared to placebo, the combined NR groups significantly improved 6-min walk distance at 3-month follow-up (+25.25 meters (90% CI: +10.43, +∞, $P = 0.015$)), maximal treadmill walking time at 6-month follow-up (+2.06 min (90% CI: + 0.24, +∞, $P = 0.075$)) and gastrocnemius muscle satellite cell abundance at 6-month follow-up (+11.14 (90% CI: +2.16, +∞, $P = 0.060$)) (Tables 4 and 5). Compared to placebo, the combined NR groups did not significantly improve 6-min walk distance (+14.05 meters, 90% CI: −1.25, +∞, $P = 0.12$), the WIQ distance score (−7.10 (90% CI:−15.19, +∞, $P = 0.87$), physical activity total activity counts/day (+10,995 (90% CI:−2,078, +∞, $P = 0.14$)), physical activity counts/min (+8.24 (90% CI:−23.85, +∞, $P = 0.37$)) or gastrocnemius muscle measures of NAD+ abundance or Percent Type I myofibers at 6-month follow-up (Tables 4 and 5).

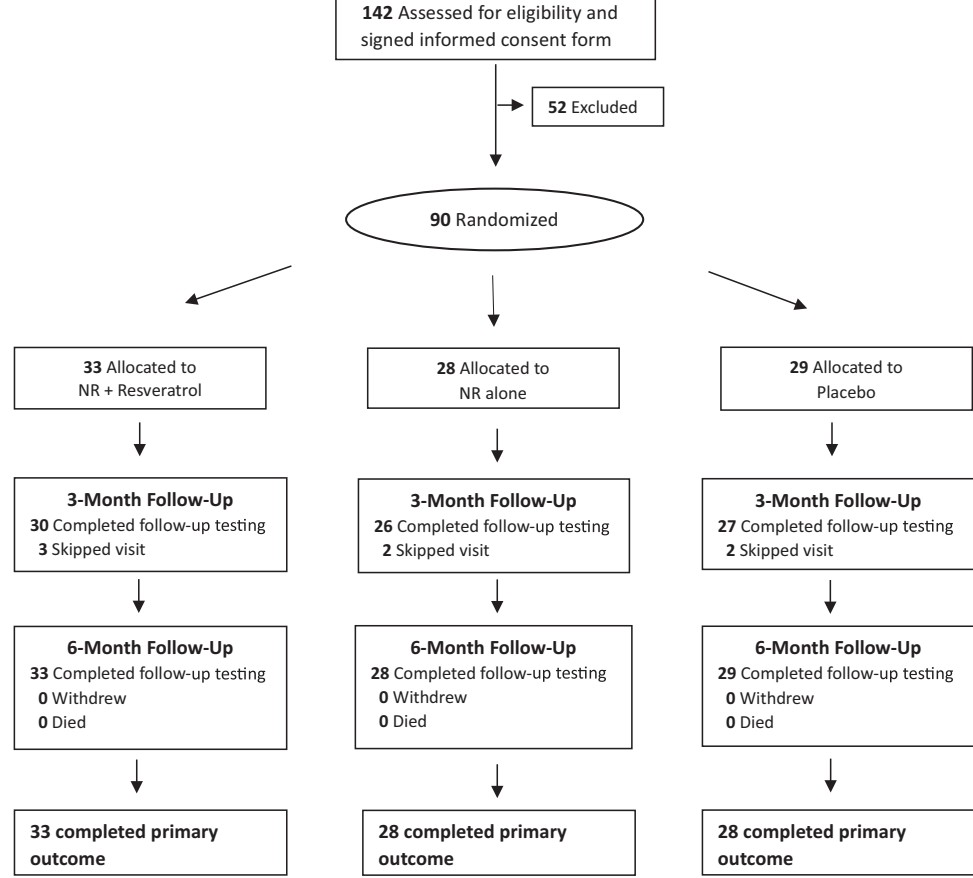

**Fig. 1 | NICE Trial consort diagram.** The figure shows the number of participants randomized to each study group and the number of people completing follow-up testing within each group.

**Table 1 | Baseline characteristics of study participants with peripheral artery disease (N = 90)**

| Baseline variable | Trial assignment | | |
|---|---|---|---|
| | NR+resveratrol (N = 33) | NR alone (N = 28) | Placebo (N = 29) |
| Age (years), mean (SD) | 70.73 (8.97) | 73.21 (9.97) | 69.72 (8.29) |
| Male, n (%) | 19 (57.58) | 16 (57.14) | 13 (44.83) |
| Female, n (%) | 14 (42.42) | 12 (42.86) | 16 (55.17) |
| African American, n (%) | 19 (57.58) | 8 (28.57) | 16 (55.17) |
| Ankle brachial index, mean (SD) | 0.67 (0.20) | 0.72 (0.18) | 0.64 (0.18) |
| Body mass index (kg/m$^2$), mean (SD) | 29.30 (6.66) | 28.65 (3.61) | 30.37 (6.39) |
| Current smoker, n (%) | 9 (27.27) | 7 (25.00) | 10 (34.48) |
| Former smoker, n (%) | 19 (57.58) | 16 (57.14) | 13 (44.83) |
| Myocardial infarction, n (%) | 6 (18.18) | 3 (10.71) | 5 (17.24) |
| Heart failure, n (%) | 3 (9.09) | 3 (10.71) | 5 (17.24) |
| Stroke, n (%) | 9 (27.27) | 1 (3.57) | 8 (27.59) |
| Angina, n (%) | 5 (15.15) | 3 (10.71) | 2 (6.90) |
| Pulmonary disease, n (%) | 8 (24.24) | 8 (28.57) | 9 (31.03) |
| Diabetes, n (%) | 15 (45.45) | 12 (42.86) | 16 (55.17) |
| Intermittent claudication, n (%) | 4 (12.12) | 4 (14.29) | 6 (20.69) |
| Atypical leg symptoms other than intermittent claudication, n (%) | 24 (72.73) | 20 (71.43) | 17 (58.62) |
| No exertional leg pain, n (%) | 5 (15.15) | 4 (14.29) | 6 (20.69) |
| 6-min walk distance (meters), mean (SD) | 336.61 (80.75) | 339.18 (103.07) | 325.60 (112.51) |
| Total treadmill walking time (min), mean (SD)$^a$ | 9.94 (5.26) | 7.29 (4.08) | 7.11 (3.44) |
| WIQ distance score, mean (SD) | 40.83 (31.66) | 45.12 (28.20) | 43.47 (24.85) |
| WIQ speed score, mean (SD) | 41.40 (25.52) | 47.01 (27.16) | 43.40 (20.50) |
| WIQ stair-climbing score, mean (SD) | 57.70 (31.32) | 56.10 (27.18) | 52.30 (30.55) |

SD standard deviation.

$^a$Sample sizes for treadmill walking time were 16 for NR alone, 17 for NR + resveratrol, and 14 for placebo. WIQ denotes walking impairment questionnaire and is scored on a 0–100 scale (100-best).

Compared to NR alone, NR + resveratrol did not significantly improve 6-min walk at 3-month follow-up or 6-min walk, maximal treadmill walking time or the WIQ distance score at 6-month follow-up (Table 6).

**Exploratory outcomes**

At 6-month follow-up, compared to placebo, NR alone significantly increased gastrocnemius muscle satellite cell abundance (+11.14 satellite cells/100 fibers, 90% CI:+2.16, +∞, P = 0.06). At 6-month follow-up, compared to placebo, there was no significant effect of NR alone or NR + resveratrol on WIQ speed score, WIQ stair-climbing score, SF-36 physical functioning score, or on gastrocnemius muscle biopsy measures of NAD+ abundance, or myofiber type (Table 7). At 6-month follow-up, compared to placebo, NR + resveratrol did not significantly increase gastrocnemius muscle satellite cell abundance (Table 7). At 6-month follow-up, compared to NR alone, NR + resveratrol did not significantly improve WIQ speed score, WIQ stair climbing score, or SF-36 physical functioning score (Supplementary Table 4). At 6-month follow-up, compared to placebo, the combined groups of NR alone and NR + resveratrol did not significantly improve the WIQ speed score, the WIQ stair

**Table 2 | Effects of nicotinamide riboside alone and nicotinamide riboside with resveratrol on the primary outcome of 6-month change in 6-min walk in peripheral artery disease**

| | Nicotinamide riboside (NR) | | | Nicotinamide riboside (NR) + resveratrol (R) | | | Placebo | | | NR vs. placebo, LSMeans (90% CI) | NR + R vs. placebo, LSMeans (90% CI) |
|---|---|---|---|---|---|---|---|---|---|---|---|
| | Baseline, mean (standard deviation) | 6-mo. follow-up, mean (standard deviation) | Within group change, LSMeans (standard error) | Baseline, mean (standard deviation) | 6-mo. follow-up, mean (standard deviation) | Within group change, LSMeans (standard error) | Baseline, mean (standard deviation) | 6-mo. follow-up, mean (standard deviation) | Within group change, LSMeans (standard error) | | |
| 6-min. walk distance (meters) | 339.2 (103.1) N=28 | 343.2 (100.0) N=28 | +7.0 (8.7) | 336.6 (80.8) N=33 | 328.9 (88.1) N=33 | -6.9 (7.8) | 325.6 (112.5) N=29 | 324.6 (89.6) N=28 | -10.6 (8.5) | +17.6 (+1.8, +∞) P=0.08 | +3.7 (-11.2, +∞) P=0.38 |

Mixed models for repeated measures (MMRM) were used to compare 6-month change in 6-min walk distance between the NR + resveratrol and placebo groups and between the NR alone and placebo groups using baseline, 3-month, and 6-month values for 6-min walk distance, adjusting for age, sex, race, and baseline 6-min walk. One participant randomized to placebo had a very low baseline value but did not complete follow-up testing, contributing to the differences between the observed mean changes in 6-min walk distance from baseline to follow-up, compared to the least square mean estimate. The a priori statistical analysis plan defined a one-sided P value of <0.10 to define statistical significance.

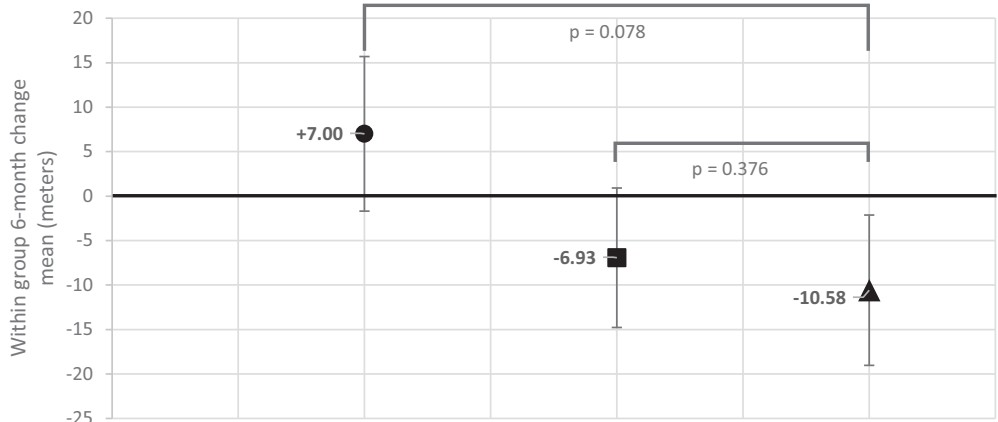

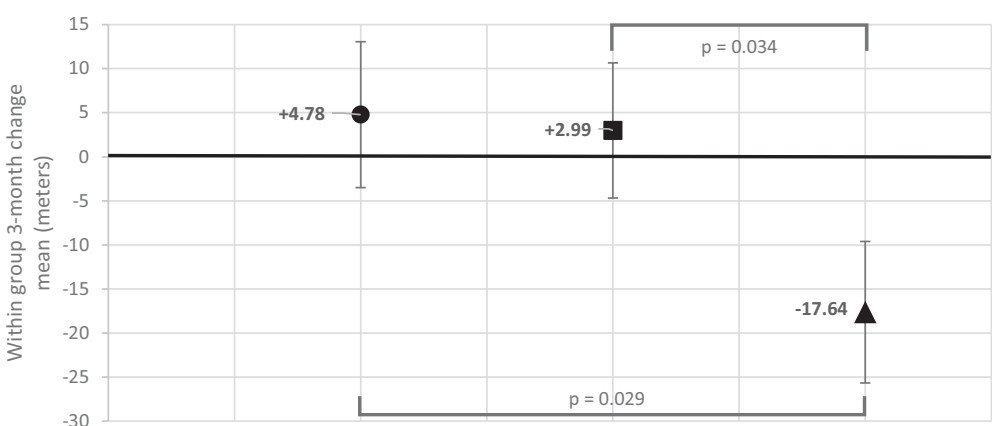

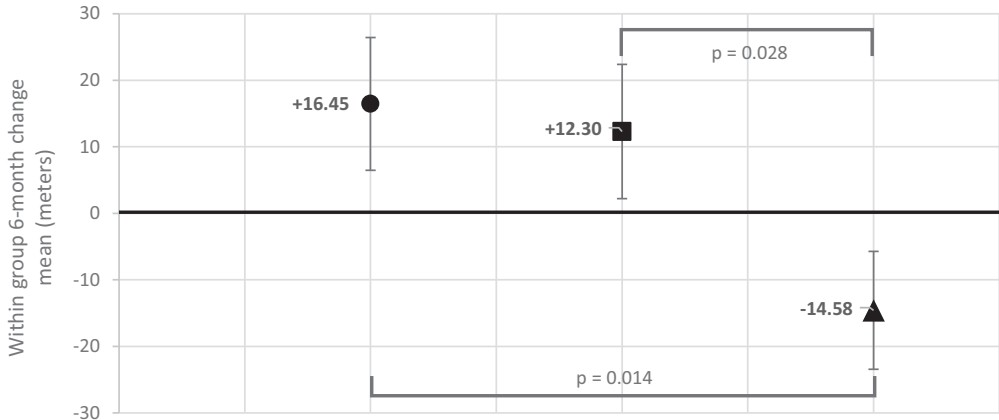

climbing score, or the SF-36 physical functioning score (Supplementary Table 3).

## Post-hoc analyses
Among all participants with 75% or greater adherence to pills, at 6-month follow-up, compared to placebo, NR improved 6-min walk by 31.0 meters (90% CI: + 13.2, +∞, P = 0.014) and NR + resveratrol improved 6-min walk by 26.9 meters (90% CI: 9.1, +∞, P = 0.028) (Fig. 2C). Among participants randomized to NR alone and among participants randomized to NR + resveratrol, those with 75% or greater adherence to study pills improved 6-min walk at 6-month follow-up, while those with <75% adherence declined in 6-min walk

**Fig. 2 | Mean change in 6-min walk distance by study group assignment.** Figure **A** depicts the 6-month change in 6-min walk (primary outcome) results for *N* = 90 participants. Mixed models for repeated measures (MMRM) were used to compare 6-month change in 6-min walk distance between the NR + resveratrol and placebo groups and between the NR alone and placebo groups using baseline, 3-month, and 6-month values for 6-min walk distance, adjusting for age, sex, race, and baseline 6-min walk. The a priori statistical analysis plan defined a one-sided *P* value of <0.10 to define statistical significance. The circle represents NR alone, the square represents NR + resveratrol, and the triangle represents placebo. Figure **B** depicts 3-month change in 6-min walk (secondary outcome) for *N* = 90 participants. Mixed models for repeated measures (MMRM) were used to compare change in 6-min walk distance between the NR + resveratrol and placebo groups and between the NR alone and placebo groups using baseline, 3-month, and 6-month values for 6-min walk distance, adjusting for age, sex, race, and baseline 6-min walk. The a priori statistical analysis plan defined a one-sided *P* value of <0.10 to define statistical significance. The circle represents NR alone, the square represents NR + resveratrol, and the triangle represents placebo. Figure **C** depicts 6-month change in 6-min walk among the 60 participants who took at least 75% of study pills. Mixed models for repeated measures (MMRM) were used to compare 6-month change in 6-min walk distance between the NR + resveratrol and placebo groups and between the NR alone and placebo groups using baseline, 3-month, and 6-month values for 6-min walk distance, adjusting for age, sex, race, and baseline 6-min walk. The a priori statistical analysis plan defined a one-sided *P* value of <0.10 to define statistical significance. The circle represents NR alone, the square represents NR + resveratrol, and the triangle represents placebo.

(Table 8). Among participants randomized to placebo, mean 6-min walk declined at 6-month follow-up, regardless of adherence (Table 8).

## Adverse events

Participants randomized to NR + resveratrol reported higher rates of diarrhea during the study (54.6%), compared to those randomized to NR alone (39.3%) and those randomized to placebo (27.6%). Participants randomized to NR + resveratrol also reported higher rates of nausea or emesis (36.4%) compared to those randomized to NR alone (14.3%) and placebo (24.1%) (Supplementary Table 5). One participant randomized to placebo experienced chest pain during the 6-min walk at three-month follow-up, was hospitalized, and later underwent coronary artery bypass grafting. No other serious adverse events were related to study participation.

## Discussion

In this double-blind randomized clinical trial of 90 participants with PAD, compared to placebo, NR improved 6-min walk distance by 17.6 meters at 6-month follow-up, consistent with a clinically meaningful effect[18,19]. Compared to placebo, NR significantly improved 6-min walk by 22.4 meters and NR + resveratrol significantly improved 6-min walk by 20.6 meters at 3-month follow-up. In post-hoc analyses, compared to placebo, NR alone significantly improved 6-min walk by 31.0 meters and NR + resveratrol improved 6-min walk by 26.9 meters among people with at least 75% adherence to study pills. NR increased satellite cell abundance in gastrocnemius muscle, compared to placebo, meeting the pre-specified criterion for statistical significance, but did not affect muscle fiber type.

To our knowledge, no prior randomized clinical trials have demonstrated beneficial effects of NR on walking performance in any human population. A clinical trial of 40 sedentary obese men showed no effect of NR 1000 mgs twice daily on changes in skeletal muscle NAD+ metabolites, insulin sensitivity, body composition, or mitochondrial activity[20–24]. Among 30 people with heart failure and ejection fraction <40%, NR 2000 mgs daily meaningfully increased plasma whole blood NAD+, but had no effect on 6-min walk, compared to placebo[24]. In contrast to participants in these prior clinical trials, PAD is characterized by lower extremity ischemia and increased lower extremity skeletal muscle reactive oxygen species (ROS)[2–4].

While the magnitude of 6-min walk improvement at 6-month follow-up was greater in the NR alone group compared to the NR + resveratrol group, the difference between these two groups was not statistically significant. Among people randomized to NR alone, 75% took 75% or more study pills, while among those randomized to NR + resveratrol, only 52% took 75% or more of study pills. The poorer adherence among those randomized to NR + resveratrol explained the smaller effect of NR + resveratrol on improved 6-min walk, compared to placebo. In analyses of participants with at least 75% adherence, NR alone and NR + resveratrol each had similarly large and clinically meaningful effects on 6-min walk. Reasons for the lower adherence rate in people randomized to NR + resveratrol are unknown, but people randomized to NR + resveratrol reported higher rates of diarrhea and higher rates of nausea or emesis, compared to those randomized to NR alone or placebo. It is possible that study intervention adherence would have been poorer and overall effect sizes from NR may have been lower if successful completion of the run-in had not been required for eligibility.

This trial has several limitations. First, the sample size was relatively small. Results require confirmation in a larger study. Second, just 17 participants had muscle biopsy at baseline and follow-up, limiting statistical power to detect NR effects on muscle. Third, due to the COVID-19 pandemic, just 26 participants had treadmill testing at baseline and follow-up. Fourth, results may not be generalizable to potential participants who would not have passed the study run-in.

Among people with PAD, NR meaningfully improved 6-min walk at 6-month follow-up, and resveratrol did not add benefit to NR alone. Among participants with at least 75% adherence, the magnitude of the effect of NR on 6-min walk was comparable to the effects of supervised exercise for PAD. Further study is needed to confirm these findings in a larger cohort of participants with PAD.

## Methods

The Institutional Review Board of Northwestern University approved the protocol. Participants gave written informed consent. The study was a parallel-design, double-blinded, randomized clinical trial conducted in Chicago at Northwestern University Feinberg School of Medicine. Enrollment occurred between 12/13/2018 and 9/14/2022. The final follow-up occurred on 4/3/2023 when the last participant completed follow-up testing. Eligible participants were randomized to one of three groups: nicotinamide riboside 1,000 mg daily, nicotinamide riboside 1000 mg + 125 mgs resveratrol daily, or placebo. Enrollment was completed when the target sample size was attained. The study protocol is available in the supplementary information file as Supplementary Note 1.

### Participant identification

Participants were recruited with advertisements on Chicago-area buses and trains, postcards mailed to people aged 50 and older, and by contacting people with PAD who previously participated in research with the principal investigator (MMM) and expressed interest in future research. Participants received $25.00 for completing 6-month follow-up testing.

### Inclusion criteria

The inclusion criterion was presence of PAD, defined by an ankle brachial index (ABI) ≤0.90 in either leg[25,26] or by medical record documentation of PAD from a vascular laboratory test obtained at a medical center or angiographic evidence of PAD (defined as 70% or

**Table 3 | Effects of nicotinamide riboside alone and nicotinamide riboside with resveratrol on secondary outcomes of physical activity, the Walking Impairment Questionnaire and objectively measured walking performance in people with peripheral artery disease**

| | Nicotinamide riboside (NR) | | | Nicotinamide riboside (NR) + resveratrol (R) | | | Placebo | | | NR vs. placebo, LSMeans (90% CI) | NR + R vs. placebo, LSMeans (90% CI) |
|---|---|---|---|---|---|---|---|---|---|---|---|
| | Baseline, mean (standard deviation) | Follow-up, mean (standard deviation) | Within group change, LSMeans (standard error) | Baseline, mean (standard deviation) | Follow-up, mean (standard deviation) | Within group change, LSMeans (standard error) | Baseline, mean (standard deviation) | Follow-up, mean (standard deviation) | Within group change, LSMeans (standard error) | | |
| WIQ distance score (0–100, 100=best) | 45.1 (28.2) N=28 | 42.9 (29.2) N=28 | -1.2 (4.4) | 40.8 (31.7) N=33 | 41.6 (30.8) N=33 | +0.82 (4.0) | 42.2 (24.4) N=28 | 48.6 (34.8) N=28 | +5.9 (4.3) | -7.1 (-15.2, +∞) P=0.87 | -5.1 (-12.6, +∞) P=0.80 |
| Physical activity – total activity counts per day | 104,869 (61,260) N=25 | 102,789 (58,695) N=25 | 4,094 (7,109) | 85,895 (33,530) N=28 | 79,588 (40,235) N=28 | -7,743 (6,478) | 86,396 (57,208) N=25 | 83,567 (53,856) N=25 | -6,901 (6,889) | 10,995 (-2,078, +∞) P=0.14 | -842 (-13,125, +∞) P=0.54 |
| Physical activity –counts per min | 486.98 (166.55) N=25 | 463.94 (140.56) N=25 | -8.81 (17.43) | 458.38 (139.66) N=28 | 448.73 (131.49) N=28 | -9.58 (16.00) | 437.44 (116.09) N=25 | 435.23 (124.84) N=25 | -17.05 (17.04) | 8.24 (-23.85, +∞) P=0.37 | 7.47 (-22.97, +∞) P=0.38 |
| 6-min. walk distance at 3-month follow-up (meters) | 339.2 (103.1) N=28 | 330.9 (99.6) N=24 | +4.8 (8.3) | 336.6 (80.8) N=33 | 340.5 (82.4) N=26 | +3.0 (7.7) | 325.6 (112.5) N=29 | 324.0 (106.2) N=24 | -17.6 (8.0) | +22.4 (7.3, +∞) P=0.029 | +20.6 (6.3, +∞) P=0.034 |
| Max treadmill walking time at 6-month follow-up (min)[a] | 7.2 (3.9) N=10 | 8.7 (3.6) N=10 | +0.9 (0.9) | 10.9 (5.0) N=8 | 10.4 (4.2) N=8 | +0.6 (1.1) | 7.1 (3.4) N=8 | 6.1 (3.7) N=8 | -1.2 (1.0) | +2.1 (0.24, +∞) P=0.08 | +1.7 (-0.21, +∞) P=0.12 |

For outcomes other than the 6-min walk test, analyses of covariance was used to compare differences between NR and placebo and between NR+ resveratrol and placebo, adjusting for age, sex, race, and the baseline value for each outcome. For the 6-min walk test, mixed models for repeated measures (MMRM) were used to compare change in 6-min walk distance between the NR alone and placebo groups and between the NR+ resveratrol and placebo groups, adjusting for age, sex, race, and baseline 6-min walk. For the 6-min walk, one participant randomized to placebo had a very low baseline value but did not complete follow-up testing, contributing to the differences between the observed mean changes in 6-min walk distance from baseline to follow-up, compared to the least square mean estimate. The a priori statistical analysis plan defined a one-sided P value of <0.10 to define statistical significance.
[a]COVID-19 restrictions resulted in a lower sample size for the treadmill testing outcome.

greater stenosis in a lower extremity artery). Vascular laboratory testing criteria acceptable for inclusion included a toe brachial index <0.70, an ABI value ≤0.90, 70% or greater stenosis in a lower extremity artery on Duplex testing, or a post-exercise ABI decline of at least 20%.

### Exclusion criteria
Exclusion criteria included above or below-knee amputation, presence of critical limb ischemia, inability to walk without a wheelchair, requiring a walker for ambulation, presence of an ulcer on the bottom of either foot, end stage kidney disease, significantly impaired liver function, and failure to pass run-in. Participants whose walking was limited by a symptom other than PAD were excluded. Potential participants who planned any major surgery or lower extremity revascularization during the next 6 months and those who completed any major surgery, coronary revascularization, or lower extremity revascularization during the previous three months were excluded. Additional exclusion criteria included supervised treadmill exercise participation during the previous 6-months, participation in a randomized clinical trial during the previous three months, unstable angina, inability to communicate in English, pregnancy or pre-menopausal state, major medical illness, life-expectancy <6 months, dementia or a mini-mental status examination score <23[27], and visual impairment limiting walking ability. People taking 250 mgs or more of nicotinamide riboside per day, vitamin B3, niacin, a slow release form of niacin, or resveratrol within the past 6 months were excluded.

### Run-in
The run-in was intended to identify and exclude potential participants who were unlikely to adhere to study interventions. Potential participants were asked to take five placebo pills daily (two NR placebo pills twice daily and one resveratrol placebo daily) for 14 days. Potential participants who did not take at least 70% of study pills were excluded.

### Randomization
A SAS program randomized eligible participants to one of three groups with equal probability (NR 1000 mgs alone, NR 1000 mgs + 125 mgs resveratrol, or placebo) using a randomly permuted block method with block sizes of four and six.

### Interventions
Interventions were administered in a double-blinded fashion for 6 months. Participants were asked to take five pills daily: Two 250 mgs of NR (or placebo) pills twice daily and one 125 mg resveratrol (or placebo) once daily. Participants were asked to record pill consumption on a paper log and return dispensed bottles at 3-month and 6-month follow-up\. Adherence was assessed using the medication log and pill counts.

### Nicotinamide riboside
NR (250 mg pills) and matching placebo were manufactured by ChromaDex.

### Resveratrol
Resveratrol capsules (125 mgs of 98% pure trans-resveratrol) and matching placebo were manufactured by ReserveAge.

### Ankle brachial index measurement
A hand-held Doppler probe (Pocket Dop II; Nicolet Biomedical Inc, Golden, CO) was used to obtain systolic pressures twice in the right and left brachial, dorsalis pedis, and posterior tibial arteries using established methods[25,26].

**Table 4 | Effects of the combined nicotinamide riboside and nicotinamide riboside + resveratrol groups, compared to placebo, in participants with peripheral artery disease**

| | NR alone and NR + resveratrol groups combined | | | Placebo | | | Comparison of change between the combined NR groups and placebo |
|---|---|---|---|---|---|---|---|
| | Baseline, mean (standard deviation) | Follow-up, mean (standard deviation) | Within group change, LSMeans (standard error) | Baseline, mean (SD) | Follow-up, mean (standard deviation) | Within group change, LSMeans (standard error) | LSMeans (90% CI) P value |
| 6-month change in 6-min walk distance (meters) | 337.79 (90.88) N = 61 | 335.47 (93.23) N = 61 | −0.24 (5.81) | 325.60 (112.51) N = 29 | 324.58 (89.63) N = 28 | −14.29 (9.97) | +14.05 (−1.25, +∞) P = 0.12 |
| 3-month change in 6-min walk distance (meters) | 337.79 (90.88) N = 61 | 335.90 (90.27) N = 50 | 3.92 (5.61) | 325.60 (112.51) N = 29 | 324.03 (106.23) N = 24 | −21.34 (9.66) | +25.25 (10.43, +∞) P = 0.015 |
| Maximal treadmill walking time (min) | 8.84 (4.64) N = 18 | 9.46 (3.82) N = 18 | 0.73 (0.64) | 7.14 (3.44) N = 8 | 6.13 (3.69) N = 8 | −1.33 (1.28) | +2.06 (0.24, +∞) P = 0.075 |
| WIQ distance score (0-100 scale, 100-best) | 42.80 (29.96) N = 61 | 42.20 (29.82) N = 61 | −0.21 (2.94) | 42.23 (24.36) N = 28 | 48.60 (34.81) N = 28 | 6.89 (5.29) | −7.10 (−15.19, +∞) P = 0.87 |
| Physical activity-total activity counts/day | 94,845 (49064) N = 53 | 90,532 (50,669) N = 53 | −1824 (4763) | 86,396 (57,208) N = 25 | 83,567 (53,856) N = 25 | −12819 (8528) | +10,995 (−2078, +∞) P = 0.14 |
| Physical activity –counts per min | 471.87 (152.11) N = 53 | 455.90 (134.74) N = 53 | −9.20 (11.77) | 437.44 (116.09) N = 25 | 435.23 (124.84) N = 25 | −17.44 (20.95) | +8.24 (−23.85, +∞) P = 0.37 |

For the 6-min walk outcomes, mixed models for repeated measures (MMRM) were used to compare change in 6-min walk distance between the combined NR and NR + resveratrol groups and the placebo group, adjusting for age, sex, race, resveratrol randomization, and baseline 6-min walk. For other outcomes, analysis of covariance (ANCOVA) was used, adjusting for baseline values for each outcome, age, sex, race, and resveratrol randomization. The a priori statistical analysis plan defined a one-sided P value of <0.10 to define statistical significance.

## Medical history

Medical history was obtained using questionnaires administered by a trained and certified health interviewer[28]. Participant's sex was designated based on the health interviewer's assessment. Race was obtained from participants using an open-ended question with fixed categories.

## Leg symptoms

Leg symptoms were characterized using the San Diego claudication questionnaire[29]. Classical intermittent claudication (IC) was defined as exertional calf pain that did not begin at rest, caused the participant to rest, and resolved within ten min of rest[29,30]. Most people with PAD do not have classic IC symptoms[30]. People with PAD who report no exertional leg symptoms (i.e. asymptomatic) and those with exertional leg symptoms other than classic IC have significantly greater functional impairment and faster functional decline than people without PAD[1,30]. To enhance generalizability of results, participants without classic IC symptoms were included in the trial[30].

## Primary outcome

The primary outcome was 6-month change in 6-min walk distance. There were two primary comparisons: Change in 6-min walk at 6-month follow-up between participants randomized to NR vs. placebo and change in 6-min walk at 6-month follow-up between participants randomized to NR + resveratrol vs. placebo.

## Secondary outcomes

Secondary outcomes were 3-month change in 6-min walk and 6-month change in maximal treadmill walking time, the Walking Impairment Questionnaire (WIQ) distance score, and daily physical activity measured by ActiGraph. Comparisons of 6-month changes in gastrocnemius muscle biopsy measures of NAD+ abundance, muscle phenotype, and satellite cell abundance between participants randomized to either NR or NR + resveratrol (combined) and placebo were secondary outcomes.

## Exploratory outcomes

Exploratory outcomes were 6-month change in the WIQ speed score, the WIQ stair climbing score, and the Short-Form-36 Physical Functioning score. Comparisons of 6-month change in gastrocnemius muscle biopsy measures of NAD+ abundance, muscle phenotype, and satellite cell abundance between NR and placebo and between NR + resveratrol and placebo were exploratory outcomes.

## 6-min walk test

Following a standardized protocol[28,31], participants walked up and down a 100-foot hallway for six min after instructions, delivered by script, to cover as much distance as possible. Distance completed after six min was recorded. In PAD, a small minimum clinically important difference (MCID) was defined as ~8 meters and a large MCID as 20 meters[18].

## Treadmill walking performance

Maximal treadmill walking time was measured using the Gardner-Skinner protocol at baseline and 6-month follow-up[32]. Treadmill testing was not available during the COVID-19 pandemic even after in-person visits became possible for other outcomes. Large MCID values for maximal treadmill walking time range from 2.5 to 4.0 min[19].

## Walking Impairment Questionnaire Scores

The WIQ is a PAD-specific measure of self-reported limitations in walking distance, speed, and stair climbing, scored on a 0–100 scale (100 = best)[33]. An MCID for the WIQ was defined as approximately five points[19].

## Health-related quality of life

The Short-Form 36 Physical Functioning (SF-36 PF) score measured health-related quality of life (range 0–100, 100 = best). The MCID is 5–7 points[34,35].

**Table 5 | Effects of the combined nicotinamide riboside and nicotinamide riboside + resveratrol groups, compared to placebo, on gastrocnemius muscle biopsy outcomes in participants with peripheral artery disease (secondary outcomes)**

| | NR alone and NR + resveratrol groups combined | | | Placebo | | | Comparison of change between the NR groups and placebo |
|---|---|---|---|---|---|---|---|
| | Baseline, mean (standard deviation) | Follow-up, mean (standard deviation) | Within group change, LSMeans (standard error) | Baseline, mean (SD) | Follow-up, mean (standard deviation) | Within group change, LSMeans (standard error) | LSMeans (90% CI) P value |
| NAD+ (pmol/mg) | 100.31 (68.33) N = 12 | 115.84 (60.95) N = 12 | 8.70 (17.92) | 92.33 (59.14) N = 5 | 105.71 (46.95) N = 5 | 43.83 (26.15) | −35.13 (−81.20, +∞) P = 0.84 |
| Satellite cell (satellite cells/ 100 fibers) | 24.72 (17.63) N = 11 | 23.50 (12.44) N = 11 | −0.81 (3.32) | 18.30 (11.21) N = 5 | 12.27 (5.66) N = 5 | −11.95 (5.18) | +11.14 (2.16, +∞) P = 0.060 |
| Type 1 myofibers (%) | 45.90 (21.73) N = 11 | 44.26 (17.77) N = 11 | −2.88 (3.58) | 56.79 (14.04) N = 5 | 50.34 (16.50) N = 5 | −8.09 (5.75) | +5.21 (−4.47, +∞) P = 0.24 |

Analyses were performed with analysis of covariance (ANCOVA), adjusting for baseline values, age, sex, race, and randomization to resveratrol.

**Table 6 | Effects of nicotinamide riboside + resveratrol, compared to nicotinamide riboside, on outcomes among people with peripheral artery disease**

| | NR + resveratrol | | | NR alone | | | Comparison of change between the NR + resveratrol vs. NR |
|---|---|---|---|---|---|---|---|
| | Baseline, mean (standard deviation) | Follow-up, mean (standard deviation) | Within group change, LSMeans (standard error) | Baseline, mean (standard deviation) | Follow-up, mean (standard deviation) | Within group change, LSMeans (standard error) | LSMeans (90% CI) P value |
| 6-month change in 6-min walk distance (meters) | 336.61 (80.75) N = 33 | 328.93 (88.14) N = 33 | −6.93 (7.84) | 339.18 (103.07) N = 28 | 343.19 (99.97) N = 28 | 7.00 (8.69) | −13.93 (−29.15, +∞) P = 0.88 |
| 3-month change in 6-min walk distance (meters) | 336.61 (80.75) N = 33 | 340.50 (82.42) N = 26 | 2.99 (7.67) | 339.18 (103.07) N = 28 | 330.92 (99.63) N = 24 | 4.78 (8.28) | −1.78 (−16.51, +∞) P = 0.56 |
| Maximal treadmill walking time (min) | 10.86 (4.95) N = 8 | 10.40 (4.17) N = 8 | 0.57 (1.11) | 7.23 (3.87) N = 10 | 8.71 (3.55) N = 10 | 0.89 (0.94) | −0.32 (−2.46, +∞) P = 0.58 |
| WIQ distance score | 40.83 (31.66) N = 33 | 41.62 (30.79) N = 33 | 0.82 (3.96) | 45.12 (28.20) N = 28 | 42.88 (29.20) N = 28 | −1.24 (4.42) | 2.06 (−5.69, +∞) P = 0.37 |
| Physical activity- total activity counts/day | 85895 (33530) N = 28 | 79588 (40235) N = 28 | −7743 (6478) | 104869 (61260) N = 25 | 102789 (58695) N = 25 | 4094 (7109) | −11837 (−24396, +∞) P = 0.89 |
| Physical activity –counts per min | 458.38 (139.66) N = 28 | 448.73 (131.49) N = 28 | −9.58 (16.00) | 486.98 (166.55) N = 25 | 463.94 (140.56) N = 25 | −8.81 (17.43) | −0.77 (−31.54, +∞) P = 0.51 |

For the 6-min walk outcome, between group comparisons at follow up visits (month 3 and month 6 visits) were based on a single mixed model for repeated measures (MMRM) analysis adjusted for baseline 6-min walk, age, sex, and race. For outcomes other than the 6-min walk, analyses were performed with analysis of covariance (ANCOVA), adjusting for baseline values, age, sex, and race. Baseline data include all randomized participants and the 3 and 6-month follow-up columns include only people who completed follow-up measurements. For the 6-min walk, one participant randomized to placebo had a very low baseline value but did not complete follow-up testing, contributing to the differences between the observed mean changes in 6-min walk distance from baseline to follow-up, compared to the least square mean estimate.

**Physical activity**

Free-living physical activity, measured as total counts per day, was acquired over seven days with the ActiGraph accelerometer[36].

**Gastrocnemius skeletal muscle biopsy**

An open muscle biopsy was performed in the medial head of the gastrocnemius muscle after subcutaneous lidocaine administration at baseline and 6-month follow-up[5–7,28]. Approximately 250 mgs of muscle tissue was removed. Samples were snap frozen in liquid nitrogen and stored at −80 degrees Celsius until testing.

**Gastrocnemius muscle measures**

For measurement of NAD+ abundance, nucleotides were measured by High Performance Liquid Chromatography (HPLC) with a Shimadzu LC-20A pump and UV-VIS detector using a Supelco LC-18-Tcolumn, normalized to muscle weight (15 cm × 4.6 cm)[37]. Satellite cell abundance and fiber type were determined on frozen muscle sections using antibodies against: Pax7 (Pax7 concentrate, DSHB; 1:100), type 1 myosin (BA.D5 concentrate, DSHB; 1:100), and laminin (Millipore-Sigma, L9393; 1:100) as previously described[5,6] (Supplementary Fig. 1).

**Other measures**

Height and weight were measured at baseline. Body mass index (BMI) was calculated as weight (kg)/[height (meters)]$^2$.

**Adverse events**

Information about adverse and serious adverse events was obtained monthly with questionnaires.

**Table 7 | Effects of nicotinamide riboside alone and nicotinamide riboside + resveratrol on exploratory muscle biopsy outcomes in people with peripheral artery disease**

| | Nicotinamide riboside (NR) | | | Nicotinamide riboside (NR) + resveratrol (R) | | | Placebo | | | NR vs. placebo, LSMeans (90% CI) | NR + R vs. placebo, LSMeans (90% CI) |
|---|---|---|---|---|---|---|---|---|---|---|---|
| | Baseline, mean (standard deviation) | Follow-up, mean (standard deviation) | Within group change, LSMeans (standard error) | Baseline, mean (standard deviation) | Follow-up, mean (standard deviation) | Within group change, LSMeans (standard error) | Baseline, mean (standard deviation) | Follow-up, mean (standard deviation) | Within group change, LSMeans (standard error) | | |
| *Exploratory outcomes* | | | | | | | | | | | |
| Satellite cells (satellite cells/ 100 fibers) | 28.04 (20.66) N = 7 | 25.76 (11.28) N = 7 | 1.73 (4.52) | 18.90 (10.56) N = 4 | 19.53 (15.11) N = 4 | −3.36 (4.33) | 18.30 (11.21) N = 5 | 12.27 (5.66) N = 5 | −9.41 (3.80) | 11.14 (2.16, +∞) P = 0.060 | 6.05 (−1.86, +∞) P = 0.16 |
| Type 1 fibers (%) | 37.94 (19.57) N = 7 | 39.80 (19.29) N = 7 | 0.17 (4.86) | 59.82 (20.04) N = 4 | 52.05 (13.45) N = 4 | −5.94 (4.88) | 56.79 (14.04) N = 5 | 50.34 (16.50) N = 5 | −5.04 (4.12) | 5.21 (−4.47, +∞) P = 0.24 | −0.90 (−9.52, +∞) P = 0.56 |
| NAD+ (pmol/mg) | 85.82 (59.93) N = 7 | 96.68 (47.85) N = 7 | −13.99 (23.44) | 120.61 (81.06) N = 5 | 142.65 (72.38) N = 5 | 31.39 (22.57) | 92.33 (59.14) N = 5 | 105.71 (46.95) N = 5 | 21.14 (20.03) | −35.13 (−81.20, +∞) P = 0.84 | 10.25 (−31.79, +∞) P = 0.37 |

Statistical comparisons were conducted using analysis of covariance (ANCOVA), adjusting for baseline values for each outcome, age, sex, and race. The a priori statistical analysis plan defined a one-sided $P$ value of <0.10 to define statistical significance.

## Sample size calculations

Anticipating a 90% follow-up rate, 30 participants randomized per group provided 80% power to detect a difference of 0.58 standard deviation (SD) of change in 6-min walk distance at a one-sided significance level of 0.10 based on a two-sample t-test[38]. Ninety participants provided more than 80% power to detect a 30-meter difference between NR + resveratrol and placebo and between NR alone and placebo at follow-up[38]. There was similar power to detect a 30-meter difference between the NR+ resveratrol group and NR alone groups. The study had 80% power to detect a between group difference of 0.58 standard deviations for other outcomes. The study had power to detect a 1.74 min difference in maximal treadmill walking time and 14.1 points for the WIQ distance score[38].

## Statistical analyses

The two primary outcomes were the differences in 6-month change in 6-min walk distance between the NR and placebo groups and between the NR + resveratrol and placebo groups. Secondary analyses included the difference in 3-month change in 6-min walk distance between the NR and placebo groups, between the NR + resveratrol and placebo groups, between the NR + resveratrol and NR groups, and between the two NR groups combined and placebo. Secondary analyses included the difference in 6-month change in 6-min walk between the NR + resveratrol and NR groups and between the two NR groups combined and placebo. Secondary analyses included differences in 6-month change in maximal treadmill walking time, the WIQ distance score, and physical activity between the NR and placebo groups, between the NR + resveratrol and placebo groups, between the NR + resveratrol and NR groups, and between the combined NR + resveratrol and NR groups and placebo. Secondary analyses included differences in 6-month change in gastrocnemius biopsy measures between the two NR groups combined and placebo. Exploratory analyses compared change in exploratory outcomes at 6-month follow-up between the groups defined for secondary analyses and 6-month changes in the gastrocnemius muscle measures between the NR + resveratrol group and placebo and between the NR and placebo groups. In post-hoc analyses, differences in 6-month changes in 6-min walk distance were compared between the NR and placebo groups and between the NR + resveratrol and placebo groups among all participants with at least 75% adherence to pills.

Pre-specified analyses were performed according to each participant's assigned group, irrespective of pill adherence. Baseline characteristics were summarized as means and standard deviations for continuous variables and frequencies and percentages for categorical variables. For the primary outcomes, mixed models for repeated measures (MMRM) were used to compare 6-month change in 6-min walk distance between the NR + resveratrol and placebo groups and between the NR alone and placebo groups using baseline, 3-month, and 6-month values for 6-min walk distance, adjusting for age, sex, race, and baseline 6-min walk. The a priori statistical analysis plan defined a one-sided $P$ value of <0.10 to define statistical significance. One-sided 90% confidence intervals (CI) were calculated for between group differences. Because the clinical trial was a Phase II trial, to help ensure that a true beneficial effect of the interventions was not dismissed due to lack of statistical significance, the pre-specified $P$ value for statistical significance was more liberal than a $P$ value of <0.05. MMRM analysis included participants missing either 3-month or 6-month follow-up visits under the missing at random assumption and results were used for baseline to 3-month follow-up and baseline to 6-month follow-up. Analyses were repeated among people with at least 75% adherent. 6-month change in 6-min walk were analyzed for each group according to whether participants had 75% or greater adherence or <75% adherence, using MMRM. For outcomes other than 6-min walk, one-sided ANCOVA tests compared changes in each outcome, adjusting for age, sex, race, and the baseline value for each outcome.

**Table 8 | Association of study drug adherence with change in 6-min walk at 6-month follow-up among participants with peripheral artery disease (*N* = 90)**

| | NR (75% were high adherers) | | NR + resveratrol (52% were high adherers) | | Placebo (76% were high adherers) | |
|---|---|---|---|---|---|---|
| | Adhered to ≥75% of study drug (*N* = 21) | Discontinued or adhered to ≤75% of drug (*N* = 7) | Adhered to ≥75% of drug (*N* = 17) | Discontinued or adhered to ≤75% of drug (*N* = 16) | Adhered to ≥75% of drug (*N* = 22) | Discontinued or adhered to ≤75% (*N* = 7) |
| 6-month change in 6-min walk (meters) | +16.5 | −13.8 | +12.3 | −25.5 | −14.6 | −3.7 |

Data shown are descriptive and no statistical testing was performed.

Analyses that compared the combined NR groups to placebo also adjusted for resveratrol randomization. There were no adjustments for multiple comparisons. Analyses were performed using SAS version 9.4.

### Reporting summary

Further information on research design is available in the Nature Portfolio Reporting Summary linked to this article.

## Data availability

The de-identified participant baseline characteristics, treatment assignments and outcome data generated in this study have been deposited in the American Heart Association GitHub database. Data can be accessed at https://doi.org/10.5281/zenodo.11099249. Data may be used for education and research purposes. The study protocol is available in the supplementary information file as Supplementary Note 1. Source data are provided with this paper.

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

## Acknowledgements

Funded by the American Heart Association, Grant Numbers 18SFRN33900097, 18SFRN33900136, and 18SFRN33970010. Nicotinamide riboside was provided by ChromaDex. Resveratrol was provided by ReserveAge

## Author contributions

M.M.M., C.R.M., K.J.D., M.H.C., L.F., P.G., J.M.G., M.R.K., K.K., D.L.J., C.A.P., R.S., L.T., L.Z., S.W. and C.L. designed the study. M.M.M., K.J.D., L.T. and L.Z. supervised data collection or statistical analyses. C.P., K.J.H., K.K., D.L.J., C.A.P., R.S. and P.Z. analyzed samples or collected samples, or interpreted data. M.M.M., C.R.M., D.Z., C.P., M.H.C., L.F., P.G., J.M.G., K.J.H., M.R.K., K.K., C.A.P., R.S., L.T., L.Z., P.Z. and L.C. interpreted results of data analyses. M.M.M. drafted the manuscript. M.M.M., C.R.M., K.J.D., D.Z., C.P., M.H.C., L.F., P.G., J.M.G., K.J.H., M.R.K., K.K., D.L.J., C.A.P., R.S., L.T., S.W., L.Z., P.Z. and C.L. provided critical feedback for the revising the manuscript. M.M.M., C.R.M., K.J.D., D.Z., C.P., M.H.C., L.F., P.G., J.M.G., K.J.H., M.R.K., K.K., D.L.J., C.A.P., R.S., L.T., S.W., L.Z., P.Z. and C.L. approved the final manuscript.

## Competing interests

Dr. McDermott reports research funding from Helixmith and other research support from ArtAssist, and Mars. The remaining authors report no competing interests.
