## [Peer Review File · Nature Communications]

REVIEWER COMMENTS

Reviewer #1 (Remarks to the Author):

Report of a double blinded RCT with three parallel arms, nicotinamide riboside 1,000 mg daily (NR), nicotinamide riboside 1,000 mg + 125 mgs or resveratrol daily (NR+resveratrol), or placebo.

The primary outcome was six month change from baseline in the six-minute walk test for NR compared with placebo and NR+resveratrol compared with placebo. There are a number of secondary outcomes including the six-minute walk test at change from baseline at 3 months.

The data for the primary outcome were analysed using a linear mixed model which takes account of the repeated measures within patients and included covariates age, sex, race and baseline six-minute walk test.

The primary outcome is reported in Table 2.

I have several questions about these results.

1. The footnote states that data are reported as mean (standard deviation). The values reported for the SD of the within group changes are too large for SD, I assume they are standard errors and this would fit in with the test results for the treatment effects.

2. The change from baseline for each treatment arm is not equal to the difference between 6month and baseline means. It is unclear what is reported as the baseline and 6 month means. Do these statistics come from the linear mixed model analysis or are they computed means with no adjustment? I would expect some slight change in the change from baseline within group when adjusted for the covariates but not the large differences reported here.

For example the difference for placebo within group change is reported as -10.6, but the difference between the 6month and baseline means is -1.0

3. The discrepancies are even greater in Table 3, the results for the 3month walk test. Once again I question whether the SD for the within group change is actually the SE.

For the NR group the difference between 3month and baseline is -8.3, and the within group change is reported as +4.8. Here even the direction of the change has altered. For placebo the difference is -1.6, but the within group change is reported as -17.6.

In table 4 it is confirmed that the SD is reported for the within group change.

Could the authors please explain what is being reported in the tables (2, 3, 4 and 5) and why we are seeing the large discrepancies that I have pointed out? It would be helpful to report the model used in each of the analyses.

Reviewer #2 (Remarks to the Author):

I read with great interest the RCT by Dr. McDermott et al titled "NICOTINAMIDE RIBOSIDE FOR PERIPHERAL ARTERY DISEASE: THE NICE RANDOMIZED CLINICAL TRIAL for consideration of publication in Nature Communications. I have the following comments I hope can help strengthen the paper.

Major Comments:

- 1) There was a significant difference in the primary comparison between NR alone and placebo at 6 months but by a $p=0.10$ criteria. I'm not certain why this is justified by the stats section - just because it was a pilot study? Also is a 2-sided t-test warranted or just a 1 sided test? th
- 2) Why did the comparison between NR+ resveratrol result in no difference for the primary 6 month endpoint? Did it negate the benefit of NR?
- 3) Can the authors expand on the enrollment criteria? Seems like patients with CLI should not be included as walking impairment is not their issue, yet this isn't listed as a criteria. How about asymptomatic disease? Later there is a comment about asymptomatic patients. Seems very odd to not focus on just those with claudication in a proof of concept pilot study.
- 4) I think the results for the comparison sin lines 255-259 should be included in the text.
- 5) I think more for the secondary/exploratory results should be included in the text/Resutls. What happened with the PROM data? What about the muscle biopsy data? I'd much rather see the intro/methods shortened and results expanded on/
- 6) I think the concluding statements should echo the exploratory nature of these findings, especially as an alpha of 0.10 was used and overall benefits were rather small.
- 7) I also think there needs to be a comment about what the authors feel resveratrol adds to this therapy. They comment on non-compliance, but do they think it adds anything? Takes away from the effect of NR? I think this is important as this was one specified arm in the trial.

REVIEWER COMMENTS

Reviewer #1 (Remarks to the Author):

1. Report of a double blinded RCT with three parallel arms, nicotinamide riboside 1,000 mg daily (NR), nicotinamide riboside 1,000 mg + 125 mgs or resveratrol daily (NR +resveratrol), or placebo. The primary outcome was six month change from baseline in the six-minute walk test for NR compared with placebo and NR+resveratrol compared with placebo. There are a number of secondary outcomes including the six-minute walk test at change from baseline at 3 months. The data for the primary outcome were analysed using a linear mixed model which takes account of the repeated measures within patients and included covariates age, sex, race and baseline six-minute walk test. The primary outcome is reported in Table 2. I have several questions about these results.

1a. The footnote states that data are reported as mean (standard deviation). The values reported for the SD of the within group changes are too large for SD , I assume they are standard errors and this would fit in with the test results for the treatment effects.

The data reported for the baseline measures were means and standard deviations (SD). As the reviewer knows, in general, standard deviations are substantially larger than standard errors. Standard errors are reported for estimating between group differences (i.e. for the least square mean estimate from the linear mixed models). The revised tables clearly indicate the standard deviations and the standard errors.

1b. The change from baseline for each treatment arm is not equal to the difference between 6month and baseline means. It is unclear what is reported as the baseline and 6 month means. Do these statistics come from the linear mixed model analysis or are they computed means with no adjustment? I would expect some slight change in the change from baseline within group when adjusted for the covariates but not the large differences reported here. For example the difference for placebo within group change is reported as -10.6, but the difference between the 6month and baseline means is -1.0

The reviewer is referring to the 6-month change in 6-minute walk distance, the primary outcome, shown in Table 2. The reason for this apparent discrepancy is that the reported mean 6-minute walk distance at baseline is based on all 90 participants. However, one participant randomized into the placebo group did not complete 6-month follow-up testing. The one participant who did not complete 6-month follow-up testing had much poorer performance on the 6-minute walk test at baseline and at 3-month follow-up, compared to the remaining 28 participants in the placebo group. Specifically, this participant had a baseline 6-minute walk distance of 113.4 meters, while the mean baseline 6-minute walk distance for the remaining 28 patients was 333.2 meters. This participant contributed to calculating the mean 6-minute walk distance at baseline, but did not contribute to calculating the mean 6-minute walk distance at month 6, resulting in a discrepancy. All between-group comparisons at follow up visits were based on the mixed model for repeated measures (MMRM) that adjusted for baseline 6-minute walk, age, sex, and race. The MMRM analysis used information from all participants but the participant missing 6-month follow-up

data for the 6-minute walk would contribute significantly less to the estimated mean change in comparison with participants who completed all follow-up tests. In this case, the one participant who did not complete 6-month follow-up testing is responsible for most of the discrepancy noted by this reviewer. For example, if we remove this participant, then the mean 6-minute walk distances at baseline and month six are 333.2 meters and 324.6 meters, respectively, suggesting a decline of 8.6 meters, which was close to the estimated mean change of -10.6 meters for the placebo group from the MMRM analysis. The remaining minor discrepancy was likely due to the adjustment of analyses for baseline six-minute walk, age, race and sex in MMRM. The data analyses have been checked multiple times and they are correct as shown. To avoid confusion, the revised manuscript tables show the baseline values, the adjusted within-group changes, and the adjusted between group changes for the comparisons. Please see our revised Tables.

2. The discrepancies are even greater in Table 3, the results for the 3month walk test. Once again I question whether the SD for the within group change is actually the SE. For the NR group the difference between 3 month and baseline is -8.3, and the within group change is reported as +4.8. Here even the direction of the change has altered. For placebo the difference is -1.6, but the within group change is reported as -17.6. In table 4 it is confirmed that the SD is reported for the within group change. Could the authors please explain what is being reported in the tables (2, 3, 4 and 5) and why we are seeing the large discrepancies that I have pointed out? It would be helpful to report the model used in each of the analyses.

Please see our response #13 above. The reason for this apparent discrepancy is that the mean 6-minute walk distance at baseline in the manuscript was based on all 90 participants. However, multiple participants did not complete 3-month follow-up testing. These participants contributed to calculating the mean 6-minute walk distance at baseline, but not the mean 6-minute walk distance at 3-months, resulting in the observed discrepancy. All between group comparisons at follow up visits (month 3 and month 6 visits) were based on a single MMRM analysis adjusted for baseline 6-minute walk, age, sex, and race. The participants who did not complete 3-month follow-up testing are responsible for most of the observed discrepancy at 3-month follow-up. For example, if we only focus on the 23 participants who completed both month 3 and month 6 follow-up testing in the placebo group, then their mean 6-minute walk distance at baseline and month 3 would be 350.3 meters and 331.5 meters, respectively, suggesting a decrease of 18.8 meters, which was close to the estimated mean change of 17.6 meters for the placebo group from the MMRM analysis. The data analyses have been double checked and they are correct as shown. To avoid confusion, the revised manuscript tables show the baseline values, the within-group changes, and the between group changes for the comparisons. Please see our revised Tables 2-4.

Reviewer #2 (Remarks to the Author):

I read with great interest the RCT by Dr. McDermott et al titled "NICOTINAMIDE RIBOSIDE FOR PERIPHERAL ARTERY DISEASE: THE NICE RANDOMIZED CLINICAL TRIAL for consideration of publication in Nature Communications. I have the following comments I hope can help strengthen the paper.

Major Comments:

3. There was a significant difference in the primary comparison between NR alone and placebo at 6 months but by a $p=0.10$ criteria. I'm not certain why this is justified by the stats section - just because it was a pilot study? Also is a 2-sided t-test warranted or just a 1 sided test?

The purpose of this Phase II clinical trial of 90 participants with PAD was to evaluate whether nicotinamide riboside, with and without resveratrol, may be effective for improving walking performance in people with PAD. The trial was designed to be an efficacy trial and was designed to detect even a small indication of efficacy. That is, if nicotinamide riboside improves walking performance, it would be important for this Phase II Trial to detect the effect. If there were no observed effect in this Phase II clinical trial, and if the Phase II clinical trial results had not met the pre-specified criteria for statistical significance, the investigation of nicotinamide riboside as a treatment for PAD would likely end with this Phase II clinical trial. Therefore, this clinical trial was designed to detect a beneficial effect, if one exists, for nicotinamide riboside in people with PAD. For this reason, the criterion for statistical significance was rather liberal, to help ensure that investigators did not miss a true benefit from the interventions. This resulted in the a priori specification of a 1-sided P value < 0.10 for criterion for statistical significance.

In response to this comment, our revised manuscript includes these statements (see page 12, first paragraph in our revised manuscript).

“Because the clinical trial was a Phase II trial, to help ensure that a true beneficial effect of the interventions was not dismissed due to lack of statistical significance, the pre-specified P value for statistical significance was more liberal than a P value of <0.05 . ”

4. Why did the comparison between NR+ resveratrol result in no difference for the primary 6 month endpoint? Did it negate the benefit of NR?

Evidence from this randomized clinical trial suggests that resveratrol did NOT negate the benefits from NR. The reason for the difference in results between the NR+ resveratrol group and the NR alone group is related to differences in the proportions of participants who attained at least 75% adherence to study drug in the NR alone group, compared to the NR + resveratrol group. Specifically, 75% of participants randomized to NR alone were at least 75% adherent to study drug, while only 52% of participants randomized to NR + resveratrol were at least 75% adherent to study drug. While it is unclear why people randomized to NR + resveratrol had poorer adherence than those randomized to NR, those randomized to NR + resveratrol had higher rates of nausea or emesis and higher rates of diarrhea, compared to people randomized to NR alone and compared to people randomized to placebo. Furthermore, participants with adherence less than 75% declined in 6-minute walk distance at 6-month follow-up, while participants with at least 75% adherence to either NR or NR + resveratrol attained meaningful and similar improvement in the 6-minute walk at 6-month follow-up (see Table B below).

Table B. Six-month change in 6-minute walk distance according to study drug adherence by group

	NR (75% were high adherers)		NR + resveratrol (52% were high adherers)		Placebo (76% were high adherers)	
	Adhered to \geq 75% of study drug (N=21)	Discontinued or adhered to \leq 75% of drug (N=7)	Adhered to \geq 75% of drug (N=17)	Discontinued or adhered to \leq 75% of drug (N=16)	Adhered to \leq 75% of drug (N=22)	Discontinued or adhered to \leq 75% (N=7)
6-month change in 6-minute walk	+16.5 meters	-13.8 meters	+12.3 meters	-25.5 meters	-14.6 meters	-3.7 meters

Our revised manuscript more clearly explains that participants randomized to NR + resveratrol were less likely to attain an adherence rate of 75% or higher. Our revised manuscript more clearly explains the reasons for the apparent poorer performance in participants randomized to NR + resveratrol.

See page 13, first paragraph, final sentence:

“Proportions of participants with at least 75% adherence to pills were 75% for NR, 52% for NR + resveratrol, and 76% for placebo.”

Please see page 15, 2nd paragraph:

“Among all participants with 75% or greater adherence to pills, at 6-month follow-up, compared to placebo, NR improved 6-minute walk by 31.0 meters (90% CI: +13.2, +∞, P=0.014) and NR + resveratrol improved 6-minute walk by 26.9 meters (90% CI: 9.1, +∞, P=0.028) (Figure 2C). Among participants randomized to NR alone and among participants randomized to NR + resveratrol, those with 75% or greater adherence to study pills improved 6-minute walk at 6-month follow-up, while those with less than 75% adherence declined in 6-minute walk (Table 5). Among participants randomized to placebo, mean 6-minute walk declined at 6-month follow-up, regardless of adherence (Table 5).”

Please also see page 16, final paragraph, and page 17, first paragraph:

“While the magnitude of 6-minute walk improvement at 6-month follow-up was greater in the NR alone group compared to the NR + resveratrol group, the difference between these two groups was not statistically significant. Among people randomized to NR alone, 75% took 75% or more study pills, while among those randomized to NR + resveratrol, only 52% took 75% or more of study pills. The poorer adherence among those randomized to NR + resveratrol explained the smaller effect of NR + resveratrol on improved 6-minute walk, compared to placebo. In analyses of participants with at least 75% adherence, NR alone and NR + resveratrol each had similarly large and clinically meaningful effects on 6-minute walk. Reasons for the lower adherence rate in people randomized to NR + resveratrol are unknown, but people randomized to NR + resveratrol reported higher rates of diarrhea during the study (54.6%), compared to those randomized to NR alone (39.3%) and those randomized to placebo (27.6%). Participants randomized to NR + resveratrol also reported higher rates of nausea or emesis (36.4%) compared to those randomized to NR alone (14.3%) and placebo (24.1%).”

5. Can the authors expand on the enrollment criteria? Seems like patients with CLI should not be included as walking impairment is not their issue, yet this isn't listed as a criteria. How about asymptomatic disease? Later there is a comment about asymptomatic patients. Seems very odd to not focus on just those with claudication in a proof of concept pilot study.

In response to this comment, the enrollment criteria have been clarified and are more thoroughly presented in our revised manuscript. Potential participants with chronic limb threatening ischemia were excluded, as indicated in our revised manuscript (see page 5 of our revised manuscript, final sentence, and page 6 (first paragraph) of our revised manuscript).

“Exclusion criteria included above or below-knee amputation, presence of critical limb ischemia, inability to walk without a wheelchair, requiring a walker for ambulation, presence of an ulcer on the bottom of either foot, end stage kidney disease, significantly impaired liver function, and failure to pass run-in. Participants whose walking was limited by a symptom other than PAD were excluded.”

Regarding the comment about inclusion of asymptomatic participants, of the 90 participants with PAD who were randomized, 15 (17%) reported no exertional leg symptoms. Participants with PAD who reported no exertional leg symptoms were not excluded because prior work shows that many of these individuals have restricted their physical activity to avoid exertional leg symptoms, that many of these individuals develop exertional leg symptoms during 6-minute walk testing, and that people with PAD who report exertional leg symptoms have significantly greater walking impairment and faster walking decline than people without PAD (McDermott MM et al. Ann Intern Med 2002;136:873-883, McDermott MM et al, JAMA 2004;292:453-461, McDermott MM et al Circulation 2000;101:1007-1012).

In response to this comment, additional analyses were performed, to evaluate results for the primary outcomes among people who reported no exertional leg symptoms (i.e. were asymptomatic) and in people who reported exertional leg symptoms.

Results are shown in Table C and indicate that effects of NR were reasonable similar (and definitely not less effective) in people with asymptomatic PAD, compared to those with symptomatic PAD and compared to the entire cohort.

Table C. Effects of nicotinamide riboside alone and nicotinamide riboside + resveratrol on the primary outcome measures in PAD participants without exertional leg symptoms

	NR group		NR + R group		Placebo group		NR vs. placebo, LSMeans (90% CI)	NR + R vs. placebo, LSMeans (90% CI)
	Baseline, Mean (SD)	Within group change, LSMeans (SE)	Baseline, Mean (SD)	Within group change, LSMeans (SE)	Baseline, Mean (SD)	Within group change, LSMeans (SE)		
6-minute walk results in asymptomatic participants	N=4 343.1 (75.1)	-0.8 (15.7)	N=5 382.5 (93.3)	-16.8 (13.6)	N=6 392.2 (63.2)	-34.7 (12.2)	33.8 (6.1, +∞) P=0.063	17.9 (-7.0, +∞) P=0.17
6-minute walk results in participants with exertional leg symptoms	N=24 338.5 (108.3)	11.8 (10.1)	N=28 328.4 (77.3)	-3.7 (8.9)	N=23 308.2 (117.0)	-2.8 (10.0)	14.6 (-3.8, +∞) P=0.15	-0.9 (-18.3, +∞) P=0.53
6-minute walk results in the entire cohort	N=28 339.2 (103.1)	7.0 (8.7)	N=33 336.6 (80.8)	-6.9 (7.8)	N=29 325.6 (112.5)	-10.6 (8.5)	17.6 (1.8, +∞) P=0.08	3.7 (-11.2, +∞) P=0.38

In response to this comment, the following sentence was added to our revised manuscript (see page eight, first paragraph, final two sentences in the paragraph):

“People with PAD who report no exertional leg symptoms (i.e. asymptomatic) and those with exertional leg symptoms other than classic IC have significantly greater functional impairment and faster functional decline than people without PAD (1,23). To enhance generalizability of results, participants without IC were included in the trial (23).”

6. I think the results for the comparisons in lines 255-259 should be included in the text.

In response, the manuscript was edited so that it includes the comparisons in the lines 255-259 from the original manuscript in the text.

See our revised manuscript (page 13, third paragraph and page 14, first and second paragraphs):

At 6-month follow-up, compared to placebo, NR significantly improved peak treadmill walking time (+2.1 minutes (90% CI: +0.24, +∞, P=0.08)), while NR + resveratrol did not significantly improve treadmill walking time (+1.7 minutes (90% CI:-0.21, +∞, P=0.12)) (Table 2). At 6-month follow-up, compared to placebo, NR alone did not significantly improve the WIQ distance score (-7.1 (90% CI:-15.2, +∞, P=0.87)), physical activity total counts/day (+10,995 (90% CI:-2,078, +∞, P=0.14)), or physical activity counts/minute (+8.24 (90% CI:-23.85, +∞, P=0.37)) (Table 2). At 6-month follow-up, compared to placebo, NR + resveratrol did not significantly improve the WIQ distance score (-5.1 (90% CI:-12.6, +∞, P=0.80)), physical activity total

counts/day (-842 (90% CI:-13,125, +∞, P=0.54)), or physical activity counts/minute (+7.47 (90% CI:-22.97, +∞, P=0.38) (Table 2).

Compared to placebo, the combined NR groups significantly improved six-minute walk distance at 3-month follow-up (+25.25 meters (90% CI: +10.43, +∞, P=0.015)) and maximal treadmill walking time (+2.06 minutes (90% CI:+0.24, +∞, P=0.075)) and satellite cell abundance (+11.14 (90% CI:+2.16, +∞, P=0.060)) (Table 3) at 6-month follow-up. Compared to placebo, the combined NR groups did not significantly improve six-minute walk distance (+14.05 meters, -1.25, +∞, P=0.12), the WIQ distance score (-7.10 (90% CI:-15.19, +∞, P=0.87), physical activity total activity counts/day (+10,995 (90% CI:-2,078, +∞, P=0.14)), physical activity counts/minute (+8.24 (90% CI:-23.85, +∞, P=0.37)) or gastrocnemius muscle measures of NAD+ abundance or Percent Type I myofibers at 6-month follow-up (Table 3).

7. I think more for the secondary/exploratory results should be included in the text/Results. What happened with the PROM data? What about the muscle biopsy data? I'd much rather see the intro/methods shortened and results expanded on.

In response, the introduction, methods, and discussion sections have been shortened. The WIQ distance score was a patient reported outcome and a secondary outcome. The WIQ speed and stair climbing scores and the SF-36 were exploratory outcomes (shown in Supplement 2 of the revised manuscript). There were three muscle biopsy outcomes that were secondary outcomes. These results are now specifically described in the revised manuscript (see page 13 final paragraph, and first two paragraphs of page 14):

“At 6-month follow-up, compared to placebo, NR alone did not significantly improve the WIQ distance score (-7.1 (90% CI:-15.2, +∞, P=0.87)), physical activity total counts/day (+10,995 (90% CI:-2,078, +∞, P=0.14)), or physical activity counts/minute (+8.24 (90% CI:-23.85, +∞, P=0.37)) (Table 2). At 6-month follow-up, compared to placebo, NR + resveratrol did not significantly improve the WIQ distance score (-5.1 (90% CI:-12.6, +∞, P=0.80), physical activity total counts/day (-842 (90% CI:-13,125, +∞, P=0.54)), or physical activity counts/minute (+7.47 (90% CI:-22.97, +∞, P=0.38) (Table 2).

Compared to placebo, the combined NR groups significantly improved six-minute walk distance at 3-month follow-up (+25.25 meters (90% CI: +10.43, +∞, P=0.015)) and maximal treadmill walking time (+2.06 minutes (90% CI:+0.24, +∞, P=0.075)) and satellite cell abundance (+11.14 (90% CI:+2.16, +∞, P=0.060)) (Table 3) at 6-month follow-up. Compared to placebo, the combined NR groups did not significantly improve six-minute walk distance (+14.05 meters, -1.25, +∞, P=0.12), the WIQ distance score (-7.10 (90% CI:-15.19, +∞, P=0.87), physical activity total activity counts/day (+10,995 (90% CI:-2,078, +∞, P=0.14)), physical activity counts/minute (+8.24 (90% CI:-23.85, +∞, P=0.37)) or gastrocnemius muscle measures of NAD+ abundance or Percent Type I myofibers at 6-month follow-up (Table 3).

Compared to NR alone, NR + resveratrol did not significantly improve six-minute walk at 3-month follow-up or six-minute walk, maximal treadmill walking time or the WIQ distance score at 6-month follow-up (Table 4).”

8. I think the concluding statements should echo the exploratory nature of these findings, especially as an alpha of 0.10 was used and overall benefits were rather small.

In response to this comment, the investigators have added a sentence to the abstract conclusion and to the manuscript conclusion as follows (see page three, final paragraph. See page 17, final paragraph):

“Further study is needed to confirm these findings in a larger clinical trial.”

9. I also think there needs to be a comment about what the authors feel resveratrol adds to this therapy. They comment on non-compliance, but do they think it adds anything? Takes away from the effect of NR? I think this is important as this was one specified arm in the trial.

We agree with this comment. The NR + resveratrol group’s performance was similar to the NR alone group for two out of the three objective measures of walking performance (6-minute walk at three month follow-up and treadmill walking time). The lack of consistency between the effects of NR alone and NR + resveratrol on 6-minute walk distance at 6-month follow-up appears to be due to the lower proportion of people randomized to NR + resveratrol who had at least 75% adherence to study drugs (52% for NR + resveratrol, compared to 75% for NR alone). This is illustrated in our revised manuscript and in the Table below:

Table D. Six-month change in 6-minute walk distance according to study drug adherence by group

	NR (75% were high adherers)		NR + resveratrol (52% were high adherers)		Placebo (76% were high adherers)	
	Adhered to ≥ 75% of study drug (N=21)	Discontinued or adhered to ≤75% of drug (N=7)	Adhered to ≥ 75% of drug (N=17)	Discontinued or adhered to ≤75% of drug (N=16)	Adhered to ≥ 75% of drug (N=22)	Discontinued or adhered to ≤ 75% (N=7)
6-month change in 6-minute walk	+16.5 meters	-13.8 meters	+12.3 meters	-25.5 meters	-14.6 meters	-3.7 meters

Our revised manuscript more clearly explains that participants randomized to NR + resveratrol were less likely to attain an adherence rate of 75% or higher. Our revised manuscript more clearly explains the reasons for the apparent poorer performance in participants randomized to NR + resveratrol.

See page 13, first paragraph, final sentence:

“Proportions of participants with at least 75% adherence to pills were 75% for NR, 52% for NR + resveratrol, and 76% for placebo.”

Please also see page 16, final paragraph, and page 17 first paragraph:

“Among people randomized to NR alone, 75% took 75% or more study pills, while among those randomized to NR + resveratrol, only 52% took 75% or more of study pills. The poorer

adherence among those randomized to NR + resveratrol explained the smaller effect of NR + resveratrol on improved 6-minute walk, compared to placebo. In analyses of participants with at least 75% adherence, NR alone and NR + resveratrol each had similarly large and clinically meaningful effects on 6-minute walk. Reasons for the lower adherence rate in people randomized to NR + resveratrol are unknown, but people randomized to NR + resveratrol reported higher rates of diarrhea during the study (54.6%), compared to those randomized to NR alone (39.3%) and those randomized to placebo (27.6%). Participants randomized to NR + resveratrol also reported higher rates of nausea or emesis (36.4%) compared to those randomized to NR alone (14.3%) and placebo (24.1%).”

REVIEWER COMMENTS

Reviewer #1 (Remarks to the Author):

Thank you for your replies to my questions and comments. Thank you for the additional documents, especially additional results and the details of the SAS programs used to analyse these clinical trial data.

The primary outcome is change from baseline for the 6 minute walk test and it is appropriate to report the baseline and 6 month measurements with the mean change from baseline. Could you please add the column of the 6 months measurements back into the tables of results, it has been removed from the revised version. All measurements should be reported with the number of observations, the mean and the standard deviation. The final column would remain the comparison of each of the two treatment arms with the placebo arm from the repeated measures analysis. This will be adjusted for the covariates and will not be identical to the mean change from baseline.

The results for the 3 month 6 minute walk test should be reported similarly. The tables of results for the secondary outcomes should be reported with baseline, 6 month and change from baseline measurements with number of observations, mean and standard deviation for each outcome. There are missing values and it is important to identify exactly where they occur.

Please label the confidence intervals correctly (90% one sided confidence interval) wherever they are reported.

Percentages with named side effects are reported in the discussion only , it would be helpful to report these in a supplementary table.

Table 4 there is a mistake in the second column heading.

Reviewer #2 (Remarks to the Author):

I appreciate the authors thoughtful responses to my comments. I review multiple manuscripts for various journals each month, and I feel that the authors did an outstanding job addressing my concerns, making important amendments to the manuscript, and overall strengthening the trial report. I applaud them for this effort and have no further comments or critiques.

REVIEWER COMMENTS

Reviewer #1 (Remarks to the Author):

1. Thank you for your replies to my questions and comments. Thank you for the additional documents, especially additional results and the details of the SAS programs used to analyze these clinical trial data.

The primary outcome is change from baseline for the 6 minute walk test and it is appropriate to report the baseline and 6 month measurements with the mean change from baseline. Could you please add the column of the 6 months measurements back into the tables of results, it has been removed from the revised version. All measurements should be reported with the number of observations, the mean and the standard deviation. The final column would remain the comparison of each of the two treatment arms with the placebo arm from the repeated measures analysis. This will be adjusted for the covariates and will not be identical to the mean change from baseline.

In response, we have added back the column showing the data for the follow-up measures. Each cell in the result tables includes the number of participants, and the mean and standard deviation. The final column continues to be the comparison of the treatment groups from the repeated measures analysis, adjusted for covariates. As noted by the reviewer, the mean change is not identical to the mean change from baseline.

2. The results for the 3 month 6 minute walk test should be reported similarly. The tables of results for the secondary outcomes should be reported with baseline, 6 month and change from baseline measurements with number of observations, mean and standard deviation for each outcome. There are missing values and it is important to identify exactly where they occur. Please label the confidence intervals correctly (90% one sided confidence interval) wherever they are reported.

These changes have also been made, including the number of observations for each value, the mean, and the standard deviation (or standard error as appropriate). The confidence intervals have been labeled correctly where they are reported.

3. Percentages with named side effects are reported in the discussion only , it would be helpful to report these in a supplementary table.

A table showing the adverse effects has been added to the supplement (see Supplement 2, Table 5). In addition, the data on the adverse events has been moved out of the discussion and placed in the final paragraph of the results in the manuscript.

Please see page 15, final paragraph, first two sentences:

Participants randomized to NR + resveratrol reported higher rates of diarrhea during the study (54.6%), compared to those randomized to NR alone (39.3%) and those randomized to placebo (27.6%). Participants randomized to NR + resveratrol also reported higher rates of nausea or emesis (36.4%) compared to those randomized to NR alone (14.3%) and placebo (24.1%) (Supplement 2, Table 5).

Please see page 17, first paragraph for revised discussion regarding adverse events:

Reasons for the lower adherence rate in people randomized to NR + resveratrol are unknown, but people randomized to NR + resveratrol reported higher rates of diarrhea and higher rates of nausea or emesis, compared to those randomized to NR alone or placebo.

4. Table 4 there is a mistake in the second column heading.

Thank you for pointing this out. This error in the column header was corrected.

Reviewer #2 (Remarks to the Author):

I appreciate the authors thoughtful responses to my comments. I review multiple manuscripts for various journals each month, and I feel that the authors did an outstanding job addressing my concerns, making important amendments to the manuscript, and overall strengthening the trial report. I applaud them for this effort and have no further comments or critiques.

We appreciate this comment from Reviewer #2.

REVIEWERS' COMMENTS

Reviewer #1 (Remarks to the Author):

I thank the authors for their replies to my comments, and for the changes that they have made to their manuscript in response. They have provided a very comprehensive account of their trial with detailed statistical results.

REVIEWER COMMENTS

Reviewer #1 (Remarks to the Author):

I thank the authors for their replies to my comments, and for the changes that they have made to their manuscript in response. They have provided a very comprehensive account of their trial with detailed statistical results.

In response, the authors thank Reviewer #1 for these comments.